

# High-throughput sequencing approach for the identification of lncRNA biomarkers in hepatocellular carcinoma and revealing the effect of ZFAS1/miR-150-5p on hepatocellular carcinoma progression

Peng Zhu[1,*], Yongyan Pei[2,*], Jian Yu[3], Wenbin Ding[1], Yun Yang[1], Fuchen Liu[1], Lei Liu[1], Jian Huang[1], Shengxian Yuan[1], Zongyan Wang[1], Fangming Gu[1], Zeya Pan[1], Jinzhong Chen[4], Jinrong Qiu[5] and Huiying Liu[5]

[1] Department of Hepatic Surgery (III), The Third Affiliated Hospital of Naval Medical University, Shanghai, China
[2] School of Chemistry and Chemical Engineering, Guangdong Pharmaceutical University, Zhongshan, Guangdong, China
[3] Department of General Surgery, The Third Affiliated Hospital of Naval Medical University, Shanghai, China
[4] State Key Laboratory of Genetic Engineering, School of Life Sciences, Fudan University, Shanghai, China
[5] Department of Biotherapy, The Third Affiliated Hospital of Naval Medical University, Shanghai, China
* These authors contributed equally to this work.

Corresponding authors
Jinrong Qiu, jrqiu@njmu.edu.cn
Huiying Liu,
liuhuiying945@163.com

## ABSTRACT

**Aims:** To screen abnormal lncRNAs and diagnostic biomarkers in the progression of hepatocellular carcinoma through high-throughput sequencing and explore the underlying mechanisms of abnormal lncRNAs in the progression of hepatocellular carcinoma.

**Methods:** The transcriptome sequencing was used to analyze the RNA expression profile and identify differentially expressed RNAs. Hub lncRNAs were screened by combining (WGCNA, ceRNA regulatory network, PPI, GO and KEGG analyses, Kaplan-Meier curve analysis, Cox analysis, risk model construction and qPCR). Thereafter, the correlation between the expression of hub lncRNAs and tumor clinicopathological parameters was analyzed, and the hub lncRNAs were analyzed by GSEA. Finally, the effects of hub RNAs on the proliferation, migration and invasion of HepG2 cells were investigated *in vitro*.

**Results:** Compared with the control group, a total of 610 lncRNAs, 2,593 mRNAs and 26 miRNAs were screened in patients with hepatocellular carcinoma. Through miRNA target prediction and WGCNA, a ceRNA was constructed, comprising 324 nodes and 621 edges. Enrichment analysis showed that mRNAs in ceRNA were involved mainly in cancer development progression. Then, the ZFAS1/miR-150-5p interaction pair was screened out by Kaplan Meier curve analysis, Cox analysis and qPCR analysis. Its expression was related to tumor stage, TNM stage and patient age. ROC curve analysis showed that it has a good predictive value for the risk of hepatocellular carcinoma. GSEA showed that ZFAS1 was also enriched in the

regulation of immune response, cell differentiation and proliferation. Loss-of-function experiments revealed that ZFAS1 inhibition could remarkably suppress HepG2 cell proliferation, migration and invasion *in vitro*. Bioinformatic analysis and luciferase reporter assays revealed that ZFAS1 directly interacted with miR-150-5p. Rescue experiments showed that a miR-150-5p inhibitor reversed the cell proliferation, migration and invasion functions of ZFAS1 knockdown *in vitro*.
**Conclusion:** ZFAS1 is associated with the malignant status and prognosis of patients with hepatocellular carcinoma, and the ZFAS1/miR-150-5p axis is involved in hepatocellular carcinoma progression.

## INTRODUCTION

Hepatocellular carcinoma is estimated to rank third in cancer-related mortality, with more than 500,000 liver cancer deaths worldwide each year (*Gao et al., 2019*; *Zhang et al., 2019*). At present, the early diagnosis of hepatocellular carcinoma can be performed by serum alpha-fetoprotein (AFP) detection, b-ultrasound and CT scanning, but the misdiagnosis rate is high (*Kulik & El-Serag, 2019*; *Zhu et al., 2019*). Despite rapid progress in the detection of hepatocellular carcinoma biomarkers, the prognosis of patients with hepatocellular carcinoma is still very poor, with a 5-year survival rate of less than 30% (*Ogunwobi et al., 2019*; *Bangaru, Marrero & Singal, 2020*), mainly because the details of the pathogenic mechanism related to hepatocellular carcinoma are not clear. Therefore, it is still very important to find more sensitive biomarkers of hepatocellular carcinoma for the diagnosis, monitoring and management of the disease.

As an important tool, high-throughput sequencing has been applied in many aspects of cancer research and treatment, including biomarker mining, cancer heterogeneity and the cancer immune microenvironment (*Jia et al., 2022*). Through high-throughput sequencing, abnormal RNAs in tumor samples can be comprehensively analyzed, which plays an important role in tumor marker screening and mechanistic research.

Current studies have shown that many lncRNAs show specific expression profiles in specific cancers, and their abnormal expression plays an important role in the occurrence and development of tumors (*Rajagopal et al., 2020*; *Tsagakis et al., 2020*; *Choudhari et al., 2020*; *Gupta et al., 2020*). Other studies have shown that lncRNAs can be stably expressed in plasma, suggesting that lncRNAs may be used as molecular biomarkers to help the diagnosis and prognosis of cancer (*Liu et al., 2019*). However, to date, most of the functions of lncRNAs are unknown, and there are no studies, including hepatocellular carcinoma. Therefore, it is necessary to mine abnormal lncRNAs in hepatocellular carcinoma through high-throughput sequencing and analyze various ways of regulating the progression of hepatocellular carcinoma. In addition, numerous studies have demonstrated that lncRNAs containing miRNA binding sites can act as competitive endogenous RNAs, and some studies have shown that lncRNAs and miRNAs interact (*Venkatesh et al., 2021*;

*Zhang et al., 2021a*). In lung carcinoma, lncRNA MALAT1 downregulation suppressed lung carcinoma progression by regulating miR-491-5p (*Dai et al., 2021*).
In retinoblastoma, lncRNA CCAT1 sponges miR-218-5p to promote EMT, cellular migration and invasion of retinoblastoma (*Meng et al., 2021*). In hepatocellular carcinoma, lncRNA ZFAS1 potentiates the development of hepatocellular carcinoma *via* the miR-624/MDK/ERK/JNK/P38 signaling pathway (*Duan et al., 2020*). However, more potential regulatory mechanisms of ZFAS1 in hepatocellular carcinoma still need to be analyzed, and the study of ZFAS1/miR-150-5p in hepatocellular carcinoma has not been reported.

In this study, we aimed to mine abnormal RNAs in hepatocellular carcinoma through high-throughput sequencing and a series of bioinformatics technologies and combined these technologies with the *in vitro* platform to study the regulatory role of key RNAs in the progression of hepatocellular carcinoma.

## MATERIALS AND METHODS

### Patients and samples

The Ethics Committee of the Third Affiliated Hospital of Naval Medical University (EHBHKY2021-K-034) approved and supervised the research project. All patients whose samples were collected gave their written informed consent. We collected 18 pairs of hepatocellular carcinomas and adjacent precancerous tissues from these patients (Material S1) (three pairs of samples for high-throughput sequencing and 15 pairs for follow-up qPCR verification). These tissues were stored at −80 °C immediately after surgical resection.

### RNA extraction and RNA sequencing

A RNeasy Mini kit (Qiagen, Hilden, Germany) was used to isolate total RNA from three hepatocellular carcinoma tissues and three adjacent precancerous tissues. Then, the concentration and purity of RNA were detected by a Nanodrop 2000. The standards of library construction were OD: A260/A280 (<2.0 and >1.8) and A260/A230 (>1.6). Then, 1% agarose gel electrophoresis (Bio-Rad, Hercules, CA, USA) was performed to analyze the RNA to ensure that it was not degraded. An Agilent 2100 Bioanalyzer (Agilent Technologies, Santa Clara, CA, USA) was used to accurately detect RNA integrity.
The RNA library was constructed using a TruSeq RNA sample preparation kit and TruSeq Small RNA (Illumina, San Diego, CA, USA). Then, Qubit 2.0 Fluorometer (Life Technologies, Carlsbad, CA, USA) and Agilent Bioanalyzer 2100 system were used to assess the library quality. After cluster generation, the library preparations were sequenced using the Illumina HiSeq 2500 platform (Illumina, San Diego, CA, USA) platform by Yuanshen Biomedical (Shanghai, China) (Novaseq 6000 SBS Kit v3-HS (200 cycles) for lncRNA and mRNA, Hiseq 2000 Truseq SBS Kit v3-HS (50 cycles) for miRNA).

### Data analysis

Finally, we obtained 150 bp paired-end sequences for each sample. To ensure the accuracy of subsequent biological information analysis, some low-quality data (a. adapter sequence; b. non-AGCT base on 5′ end; c. containing N bases and quality score less than 10%; d.

reads that were too short (<18 nt); e. read ends with low sequencing quality) in the original sequencing data were first removed using Fastx-Toolkit (Version 0.0.13, default parameters (http://hannonlab.cshl.edu/fastx_toolkit/)) to obtain high-quality clean data. Next, the clean data were compared with the human genome (GRCh38) using HISAT2 (*Lachmann et al., 2020*). To study and mine the transcriptome data more comprehensively, we used StringTie (*Pertea et al., 2015*) to perform transcriptome data assembly based on alignment results and identify new genes and transcripts. RefSeq (http://www.ncbi.nlm.nih.gov/refseq/) and Ensembl transcript databases (https://www.ensembl.org) were selected as references for mRNA and lncRNA comments. miRNA used the Rfam database (V11.0, https://rfam.org/) for annotations. The Coding Potential Assessment Tool (CPAT, https://bio.tools/cpat) was used to predict lncRNAs and mRNAs, and BLAST (v 2.3.0, http://blast.ncbi.nlm.nih.gov/) was used to predict miRNAs. The read count of each text is normalized and normalized to the length of a single text and the total read count mapped in each sample, which is expressed as FPKM. Then, the R package limma was utilized to identify differentially expressed genes (DEGs) and differentially expressed miRNAs (DEMs) with the criteria : In the analysis, we used |log2 (fold-change)|>1, False Discovery Rates (FDR) <0.05 and $p < 0.05$.

## Weighted gene coexpression network analysis (WGCNA)

First, lncRNA expression profiles, miRNA expression profiles, and mRNA expression profiles from hepatocellular carcinoma were downloaded from TCGA, and unqualified RNAs were removed. Then, based on the scale-free topology criterion, the weighted adjacency matrix (WAM) was established by selecting the appropriate soft threshold (β) by the pick-soft threshold. The hub correlation threshold was set to 0.9, the minimum number of genes of modules was 20, and the threshold of module merging was 0.25. According to the topological overlap matrix (TOM) function, the adjacency matrix was transformed into TOM, and the RNAs that were closely related to hepatocellular carcinoma patients and highly co-expressed were classified into the same module.

## ceRNA network analysis

Starbase 2.0 (http://starbase.sysu.edu.cn/) was used to determine the possible target mRNAs and lncRNAs of maladjusted miRNAs. Combined with the core RNAs screened by WGCNA, overlapping RNAs were obtained. Then, Cytoscape 3.9 was used to draw the ceRNA network with overlapping RNAs.

## GO and KEGG pathway analysis and gene set enrichment analysis

Gene Ontology (GO) and Kyoto Encyclopedia of Genes and Genomes (KEGG) enrichment analyses of mRNAs in ceRNA were carried out by using Metascape (https://metascape.org/). The Q value is used to test the reliability of the analysis. The enrichment analysis circle diagram was drawn by Sangerbox 3.0 (http://sangerbox.com/). Gene set enrichment analysis (GSEA) was carried out using normalized RNA-Seq data in TCGA-LIHC. The number of permutations is set to 100. The GO and KEGG pathways of

**Table 1 The sequences of primers.**

| Gene symbol | Primer | Sequences (5′-3′) |
| --- | --- | --- |
| ZFAS1 | Forward primer | GAAGAGGGAGTCACCACTG |
| | Reverse primer | TTGGCCAACAATAAACTCGT |
| FBXL19-AS1 | Forward primer | CCCTTCCCTCTGTCTTCTG |
| | Reverse primer | ACCACAACCAACAAGTCCT |
| β-actin | Forward primer | ACTGGGACGACATGGAGAAAA |
| | Reverse primer | TGGCTGGGGTGTTGAAGG |

lncRNAs were further analyzed by GSEA to explore the possible biological functions of lncRNAs ($p < 0.05$).

## Protein-protein interaction (PPI) network

Build a PPI network with mRNAs in ceRNA. PPI was performed using STRING database 11.5 (https://string-db.org/) and a comprehensive score >0.5 as the critical value. In addition, the PPI network was built and visualized using Cytoscape 3.9.

## Kaplan-Meier survival analysis of lncRNAs

To explore the predictive value of lncRNA expression levels on the survival of patients with hepatocellular carcinoma, GEPIA 2.0 (http://gepia2.cancer-pku.cn/) was used for Kaplan-Meier survival analysis. The statistical significance was $p < 0.05$. Then, StarBase 2.0 (http://starbase.sysu.edu.cn/) was used to rescreen the analyzed data.

## Construction of the lncRNA Cox proportional risk regression model

Based on the data of univariate and multivariate Cox proportional hazards regression analyses, we constructed a Cox proportional hazards regression model and obtained the formula (b exp ($lncRNA_1$) + b exp ($lncRNA_2$) +…+ b exp ($lncRNA_n$)), where b represents the multivariate Cox regression coefficient and exp () represents the expression level of prognostic lncRNAs. Then, we calculated the survival rate of the high-risk group and the low-risk group and drew the survival receiver operating characteristic (ROC) curve of 1-year, 3-year, and 5-year subjects to test the feasibility of the predictive ability of the model.

## Real time quantitative PCR

qRT-PCR analysis was performed on 15 pairs of hepatocellular carcinoma clinical experimental samples. Total RNA was extracted from the samples as described above. A Revert Aid First Strand cDNA Synthesis kit (Thermo Fisher Scientific, Waltham, MA, USA) was used to reverse-transcribe the RNA. cDNA was analyzed using 2×SYBR Green qPCR Master mix (Thermo Fisher Scientific, Waltham, MA, USA). The primer sequences are shown in Table 1. The lncRNA primer was synthesized by Sangon (Shanghai, China), and the miRNA primer was purchased from RiboBio (Guangzhou, China). lncRNA used

β-actin as an internal control, and miRNA used U6 as an internal control. The relative expression of the target gene was calculated with the $2^{-\Delta\Delta Ct}$ method.

## Cell culture and transfection

HepG2 cells were purchased from Beijing Beina Chuanglian Biotechnology Research Institute and cultured in Dulbecco's modified Eagle's medium (DMEM, with 10% fetal bovine serum, 100 U/ml penicillin and 100 mg/ml streptomycin, Gibco, Billings, MT, USA) at 37 °C in a 5% $CO_2$ incubator. HepG2 cells were transfected with 50 nM siRNA (Ribobio, Guangzhou, China), 2 μg overexpression vector (Ribobio, Guangzhou, China) and 50 nM miRNA mimic (Ribobio, Guangzhou, China) and NC control using Lipofectamine 3000 reagent (Invitrogen, Shanghai, China) according to the manufacturer's instructions.

## Cell counting kit-8 (CCK-8) assays

CCK-8 (Sigma Aldrich, St. Louis, MI, USA) was used to detect the proliferation of HepG2 cells in 96-well plates. A total of 5,000 cells were added to each well and incubated for 24, 48 and 72 h. Then, the cells were incubated with CCK-8 reagent (10 μl per well) for 2 h, and the absorbance at OD 450 was detected with a microplate reader (Thermo Fisher Scientific, Waltham, MA, USA). Each experiment was performed three times.

## Soft agar assay

HepG2 cells were inoculated in six-well plates ($3 \times 10^5$ cells/well). Forty-eight hours after transfection, the cells were collected and counted. The cell density was adjusted to 2,000 cells/ml with medium containing 10% fetal bovine serum. The cells were fully mixed with medium containing 0.7% agarose and quickly placed on the solidified layer of medium containing 1.2% agarose. Then, 500 μl of medium was added every 3 d. After 7 d of culture, the cells were fixed with 4% paraformaldehyde, stained with 0.005% crystal violet and photographed under a microscope. We calculated the field of vision with the number of clones greater than 0.05 mm and all clones, and clone formation rate = the number of clones greater than 0.05 mm/the number of all clones × 100%.

## Wound-healing assay

Collect HepG2 cells in logarithmic phase and seeded in 96-well plates at $4 \times 10^5$ cells/well. After 24 h, the fused cell was scratched with a 200 ml sterile pipette, and the suspended cells were removed with phosphate buffer saline (PBS, Beyotime, Jiangsu, China). The scratches were observed with a microscope. ImageJ software was used to evaluate the migration rate.

## Transwell assay

After transfection, HepG2 cells seeded in upper chamber at $2 \times 10^3$ cells/well, and mixed with Matrigel. Then DMEM containing 20% fetal bovine serum was added to the lower layer of the chamber. After 24 h, the upper chamber cells were fixed with paraformaldehyde (4%, Sigma Aldrich, St. Louis, MO, USA) and stained with crystal violet

(1%, Sigma Aldrich, St. Louis, MO, USA). The light microscope (Olympus, Shinjuku City, Japan) was used to observe and count cells.

## Raw sequencing data

The raw sequencing data of this study have been uploaded to GEO; lncRNA and mRNA sequencing data can be found here: https://www.ncbi.nlm.nih.gov/geo/query/acc.cgi?acc=GSE185799; miRNA sequencing data were provided by Yongyan Pei, which can be found here: https://www.ncbi.nlm.nih.gov/geo/query/acc.cgi?acc=GSE185913.

## Statistical analysis

GraphPad 8 and Excel 2019 were used to analyze the experimental data. The difference was statistically significant ($p < 0.05$). All the data are presented as the mean ± standard deviation. Three technical replicates were collected from each independent experiment for qPCR. A $t$-test was used to evaluate the significant difference between two groups. The significance among multiple groups was determined by one-way analysis of variance (ANOVA).

# RESULT

## Results of high-throughput sequencing and overview of transcripts

To acquire a comprehensive and in-depth understanding of RNA (lncRNA, mRNA, miRNA) transcripts in hepatocellular carcinoma, we carried out high-throughput RNA sequencing on tissue samples from three patients with hepatocellular carcinoma and three adjacent tissues of hepatocellular carcinoma (Figs. 1A and 1B). In total, we obtained 67.62 million raw reads for lncRNAs and mRNAs and 7.29 million raw reads for miRNAs. The sequencing Q30 of each sample was >90%, which indicated good sequencing quality (Material S2). Ultimately, 67.58 million clean reads and 6.84 million clean reads were acquired after screening for lncRNAs/mRNAs and miRNAs, respectively. Furthermore, to confirm the authenticity and validity of the RNA-seq data, we performed Pearson correlation coefficient analysis on transcript expression levels among different samples (Fig. 1C). Principal component analysis (PCA) showed that the expression characteristics of RNAs (lncRNAs, miRNAs and mRNAs) were different between hepatocellular carcinoma patients and controls (Fig. 1D). The results showed that the tumor tissues or the control tissues themselves showed a high correlation. Then, all genes and transcripts were annotated with multiple databases (NR, SwissProt, PFAM, GO, KEGG, and STRING) based on the reference genome.

## Identification of differentially expressed lncRNAs, mRNAs and miRNAs between hepatocellular carcinoma tissues and adjacent tissues

In this study, differentially expressed RNAs (lncRNAs, miRNAs and mRNAs) between hepatocellular carcinoma tissues and adjacent tissues were analyzed by R. By high-throughput sequencing analysis, a total of 10,075 lncRNAs, 15,021 mRNAs and 1,192 miRNAs were obtained (Materials S3–S5). Furthermore, among the total expressed RNAs (lncRNAs,

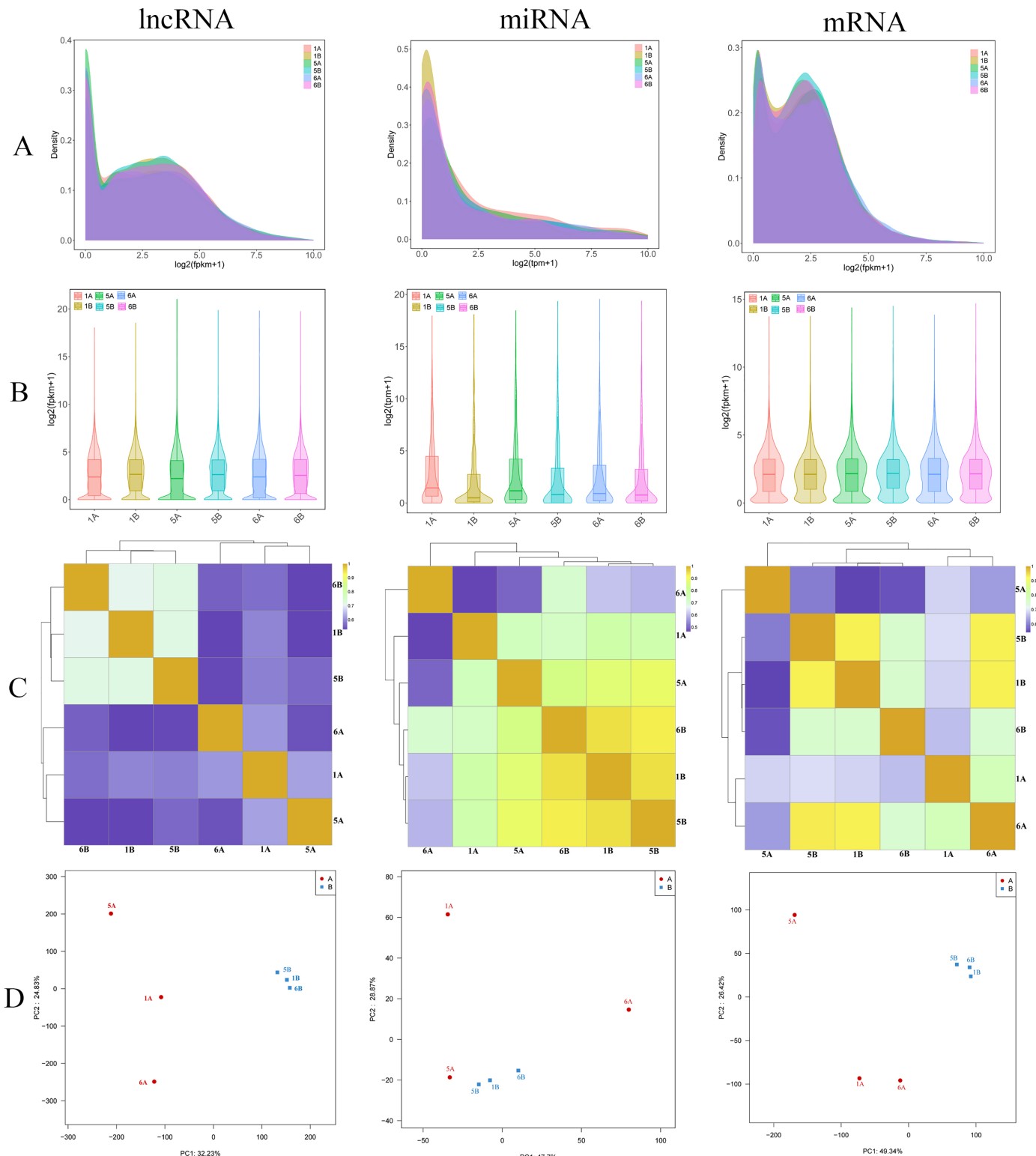

**Figure 1 Sequencing data analysis.** (A and B) The lncRNA, miRNA and mRNA expression density distribution graphs and violin graphs of the sequenced samples. (C) Sample correlation heatmap based on lncRNA, miRNA and mRNA expression of the sequenced samples. (D) PCA of lncRNAs, miRNAs and mRNAs of the sequenced samples. Group A (1A, 5A and 6A) represents the tumor group, and Group B (1B, 5B and 6B) represents the normal group.

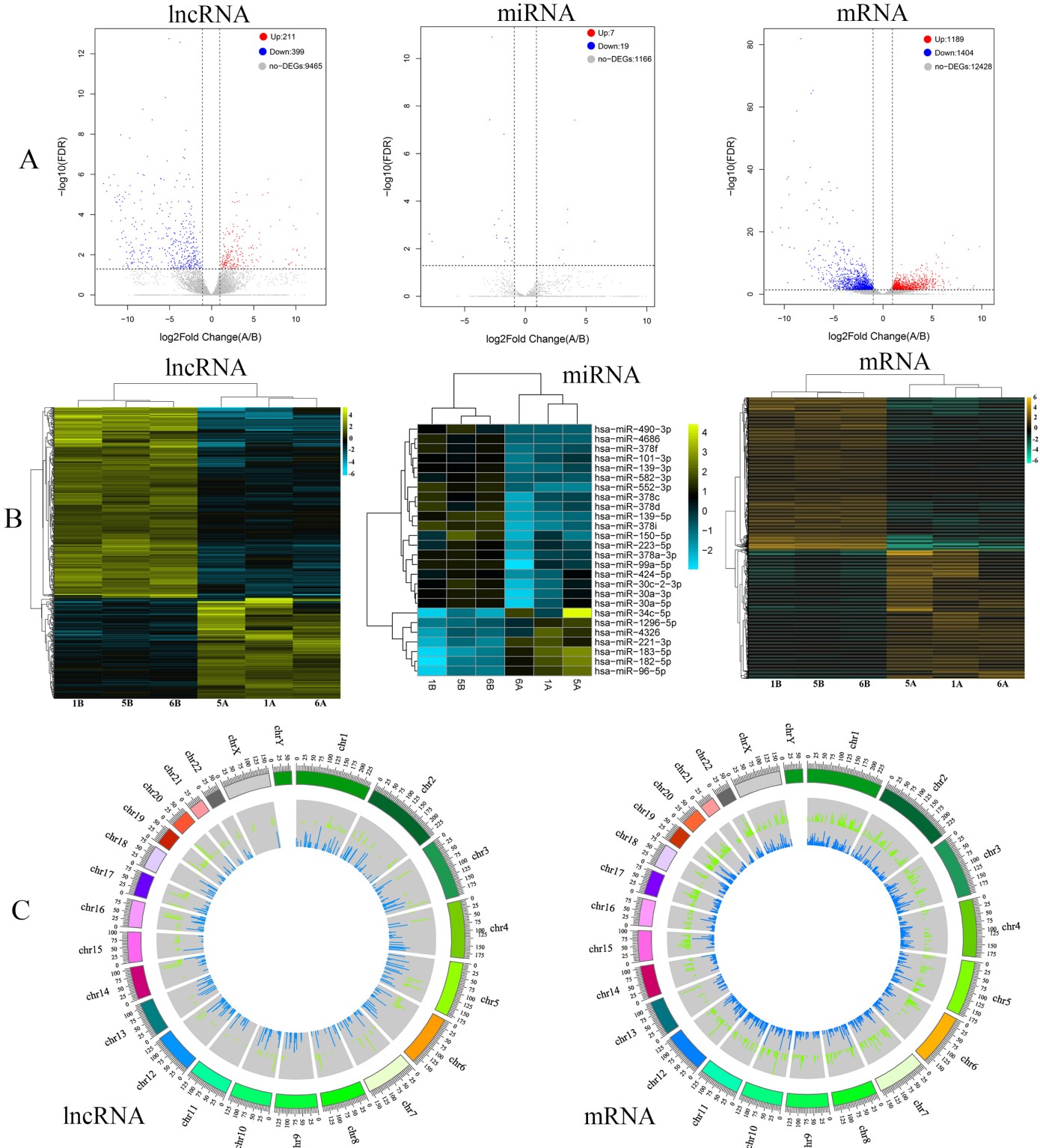

**Figure 2 Analysis of differentially expressed RNAs.** (A) Volcano plots of differentially expressed lncRNAs, miRNAs and mRNAs in sequencing data of hepatocellular carcinoma. The red dots represent upregulated RNAs, and the blue dots represent downregulated RNAs. The gray dots represent genes with no significant difference. (B) Heatmap of differentially expressed lncRNAs, miRNAs and mRNAs in the sequencing data of hepatocellular carcinoma. (C) Circos plots of lncRNAs and mRNAs of sequencing data in the human genome (hg19). The outer tracks represent the
**Figure 2 (continued)**
chromosome. For the two tracks, the outer one (red) represents the levels of upregulated lncRNAs and mRNAs between hepatocellular carcinoma and precancer tissues, and the inner one (green) represents the levels of downregulated lncRNAs and mRNAs between hepatocellular carcinoma and precancer tissues.                                               

miRNAs and mRNAs), 610 lncRNAs were significantly differentially expressed, of which 211 were upregulated and 399 were downregulated (Fig. 2A). In addition, 2,593 mRNAs were significantly differentially expressed; 1,189 were upregulated and 1,404 were downregulated (Fig. 2A). Additionally, we obtained 26 miRNAs, among which seven were upregulated and 19 were downregulated (Fig. 2A). The top 10 ranked upregulated and downregulated differentially expressed RNAs (lncRNAs, mRNAs and miRNAs) are shown in Tables 2–4, respectively. To further understand the different characteristics between hepatocellular carcinoma tissues and adjacent tissues, cluster analysis was performed for the differentially expressed lncRNAs, miRNAs and mRNAs. The heatmap showed that the RNAs (lncRNAs, miRNAs and mRNAs) in the hepatocellular carcinoma samples and adjacent samples were well distinguished (Fig. 2B). Moreover, the distribution of differentially expressed lncRNAs and mRNAs on each chromosome is shown in Fig. 2C.

## Weighed gene coexpression network analysis (WGCNA) of RNAs (lncRNAs, miRNAs and mRNAs) in hepatocellular carcinoma

RNA and miRNA expression profile was downloaded (TCGA-LIHC). Then, low-expression genes (median absolute deviation greater than 50%) and outlier samples were filtered out, and the processed expression profiles were analyzed by gene coexpression network analysis (WGCNA). As shown in Fig. 3A, when $\beta = 3$ and $R^2 = 0.97$, all lncRNAs after screening were clustered into 15 modules (Fig. 3D). The same treatment was performed on miRNAs and mRNAs. For miRNAs and mRNAs, the most suitable power values were 2 ($R^2 = 0.94$) and 4 ($R^2 = 0.90$) (Figs. 3B and 3C). The dendrogram plot of miRNAs and mRNAs is shown in Figs. 3E and 3F. miRNAs were clustered into 13 modules, and mRNAs were clustered into 39 modules.

To further understand the correlation between modules and hepatocellular carcinoma, we obtained the correlation between module eigengenes and sample traits, including cancer, age, sex, survival, T stage, N stage, M stage and stage. For lncRNAs, we found that the blue module eigengene containing 15 hub lncRNAs had the highest correlation index with traits ($r = 0.81$, $p = 0.03$, Fig. 3G). Additionally, we found that the green module eigengene containing 39 hub miRNAs had the highest negative correlation index with traits ($r = -0.72$, $p = 4.5E-70$, Fig. 3H). Additionally, 797 mRNAs were contained in the dark orange module eigengene with the highest correlation index with traits ($r = -0.78$, $p = 6.9E-88$, Fig. 3I). Thus, we obtained hub modules for RNAs in hepatocellular carcinoma by WGCNA.

## Construction of the lncRNA-miRNA-mRNA regulatory network

To clarify the relationship between differentially expressed RNAs (lncRNAs, miRNAs and mRNAs), we established a lncRNA-miRNA-mRNA ceRNA network. First, we predicted the potential targeting lncRNAs and mRNAs of miRNA through starBase. Then, we
**Table 2 Differential expression lncRNAs.**

| ID | Symbol | lncRNA type | logFC | $p_{adj}$ | Chr | $p$ value |
|---|---|---|---|---|---|---|
| **Up regulation** | | | | | | |
| ENSG00000226644 | AL121899.1 | lincRNA | 12.58588 | 9.21E−05 | 20 | 8.06E−07 |
| ENSG00000228952 | LINC02041 | lincRNA | 11.12840 | 0.013248 | 3 | 0.00051 |
| ENSG00000283627 | AL137785.1 | lincRNA | 11.08179 | 0.004581 | 14 | 0.00011 |
| ENSG00000282381 | AC104073.4 | lincRNA | 10.76832 | 0.033715 | 7 | 0.00188 |
| ENSG00000279621 | AC020978.8 | TEC | 10.61554 | 1.86E−06 | 16 | 5.32E−09 |
| ENSG00000261633 | AC018552.3 | antisense | 10.39446 | 0.010659 | 16 | 0.00037 |
| ENSG00000251461 | AC091133.3 | antisense | 10.18125 | 0.005531 | 17 | 0.00015 |
| ENSG00000259106 | AC008056.2 | antisense | 10.14539 | 0.006568 | 14 | 0.00019 |
| ENSG00000230506 | AL354824.1 | lincRNA | 10.01828 | 0.033593 | 20 | 0.00187 |
| ENSG00000251350 | LINC02475 | lincRNA | 10.01675 | 0.000535 | 4 | 7.33E−06 |
| **Down regulation** | | | | | | |
| ENSG00000248359 | AC010280.1 | lincRNA | −12.8558 | 2.90E−06 | 5 | 2.90E−06 |
| ENSG00000230921 | HAO2-IT1 | sense_intronic | −12.4964 | 7.07E−06 | 1 | 7.07E−06 |
| ENSG00000257737 | AC025265.2 | antisense | −12.2028 | 1.42E−06 | 12 | 1.42E−06 |
| ENSG00000269888 | AC112491.1 | lincRNA | −12.1293 | 0.016964 | 3 | 0.016964 |
| ENSG00000248884 | AC010280.2 | lincRNA | −12.0662 | 3.54E−06 | 5 | 3.54E−06 |
| ENSG00000251443 | LINC02160 | lincRNA | −11.8975 | 0.000213 | 5 | 0.000213 |
| ENSG00000233590 | AC016395.1 | lincRNA | −11.6254 | 1.03E−06 | 10 | 1.03E−06 |
| ENSG00000250266 | LINC01612 | lincRNA | −11.4874 | 1.41E−05 | 4 | 1.41E−05 |
| ENSG00000259037 | BX927359.1 | antisense | −11.3504 | 2.38E−05 | 14 | 2.38E−05 |
| ENSG00000253939 | AC007991.3 | antisense | −11.2575 | 2.48E−05 | 8 | 2.48E−05 |

Note:
Top 10 (up- and downregulated) differentially expressed lncRNAs in normal tissues and hepatocellular carcinoma tissues.

compared the predicted RNAs with differentially expressed RNAs and hub RNAs from WGCNA. Ultimately, a total of eight lncRNAs, nine miRNAs and 307 mRNAs (Material S6) were involved in constructing the ceRNA network using Cytoscape, which contained 324 nodes and 621 edges (Fig. 4).

## GO and KEGG analyses and protein-protein interaction (PPI) network construction of mRNAs in the ceRNA network

To better understand the function of mRNA in the ceRNA network, we used Metasscape and Sanger box to carry out gene ontology (GO) functional analysis and Kyoto encyclopedia of genes and genomes (KEGG) analysis. The results of GO and KEGG enrichment analyses are presented in Figs. 5A and 5B. These mRNAs were mainly enriched in GTPase-mediated signal transduction, cell differentiation, leukocyte differentiation, neuron projection morphogenesis, regulation of cell projection organization, cell proliferation, cell-matrix adhesion, inflammatory response, negative regulation of intracellular signal transduction and myeloid leukocyte differentiation. KEGG pathway enrichment results showed that mRNAs were mainly enriched in several pathways, including the VEGFA-VEGFR2 signaling pathway, Ras signaling pathway,

**Table 3 Differential expression mRNAs.**

| ID | Symbol | gene type | logFC | $p_{adj}$ | Chr | $p$ value |
|---|---|---|---|---|---|---|
| **Up regulation** | | | | | | |
| ENSG00000107984 | DKK1 | pc | 9.760657 | 5.69E−16 | 10 | 5.69E−16 |
| ENSG00000139219 | COL2A1 | pc | 9.207937 | 0.001731 | 12 | 0.001731 |
| ENSG00000285304 | Z83844.3 | pc | 9.121696 | 0.009745 | 22 | 0.009745 |
| ENSG00000043355 | ZIC2 | pc | 8.648202 | 4.32E−15 | 13 | 4.32E−15 |
| ENSG00000206557 | TRIM71 | pc | 7.671976 | 0.01305 | 3 | 0.01305 |
| ENSG00000242265 | PEG10 | pc | 7.480217 | 7.01E−06 | 7 | 7.01E−06 |
| ENSG00000147257 | GPC3 | pc | 7.067548 | 1.53E−19 | X | 1.53E−19 |
| ENSG00000156564 | LRFN2 | pc | 6.828666 | 0.000965 | 6 | 0.000965 |
| ENSG00000112164 | GLP1R | pc | 6.675201 | 0.000107 | 6 | 0.000107 |
| ENSG00000159217 | IGF2BP1 | pc | 6.496931 | 3.18E−09 | 17 | 3.18E−09 |
| **Down regulation** | | | | | | |
| ENSG00000156006 | NAT2 | pc | −11.19968226 | 1.36E−20 | 8 | 2.59E−23 |
| ENSG00000263761 | GDF2 | pc | −10.37224844 | 4.83E−22 | 10 | 8.56E−25 |
| ENSG00000140505 | CYP1A2 | pc | −10.28584106 | 2.05E−28 | 15 | 2.49E−31 |
| ENSG00000180745 | CLRN3 | pc | −10.2196129 | 2.06E−16 | 10 | 5.18E−19 |
| ENSG00000019169 | MARCO | pc | −9.718074611 | 1.19E−37 | 2 | 6.50E−41 |
| ENSG00002255974 | CYP2A6 | pc | −9.652187815 | 0.000200358 | 19 | 8.65E−06 |
| ENSG00000182566 | CLEC4G | pc | −9.594142896 | 2.22E−38 | 19 | 1.06E−41 |
| ENSG00000104938 | CLEC4M | pc | −9.558825034 | 6.64E−22 | 19 | 1.22E−24 |
| ENSG00000145824 | CXCL14 | pc | −9.501511504 | 1.13E−32 | 5 | 1.08E−35 |
| ENSG00000264006 | AKR1C8P | pc | −9.432829243 | 9.18E−16 | 10 | 2.69E−18 |

**Note:**
Top 10 (up- and downregulated) differentially expressed mRNAs in normal tissues and hepatocellular carcinoma tissues.

oxytocin signaling pathway, G protein signaling pathways, proteoglycans in cancer, PI3K-Akt signaling pathway, IL-18 signaling pathway, signaling by interleukins, p53 signaling pathway, TCR signaling pathway, NF-kappa B signaling pathway and signaling by Rho GTPases. The GO and KEGG analyses indicated that biological processes and immune-related molecular mechanisms play a role in hepatocellular carcinoma.

Moreover, we analyzed the interactions of 307 mRNAs in the ceRNA network and constructed a PPI network. The PPI networks included 176 nodes and 396 edges (Fig. 6A). Then, we further filtered the PPI network through the MCODE plug-in and obtained a subnetwork (Fig. 6B). The subnetwork consisted of 12 hub mRNAs: JAK2 (degree = 17), CCR7 (degree = 22), ESR1 (degree = 18), CXCL12 (degree = 22), CD274 (degree = 18), CSF1 (degree = 15), PTGS2 (degree = 18), IRF4 (degree = 15), TLR4 (degree = 26), TLR1 (degree = 16), CCL22 (degree = 9), and CD163 (degree = 9). By comparing the mRNAs in the ceRNA subnetwork with the mRNAs in the PPI subnetwork, three common mRNAs (ESR1, CXCL12 and IRF4) were obtained, which should play an important role in the development of hepatocellular carcinoma and used as follow-up research objects.

**Table 4 Differential expression miRNAs.**

| MiRNA_id | log2FC (A/B) | $p_{val}$ | $p_{adj}$ |
|---|---|---|---|
| **Up regulation** | | | |
| hsa-miR-183-5p | 4.06726107 | 2.22E−10 | 3.87E−08 |
| hsa-miR-182-5p | 3.477582683 | 2.14E−06 | 0.000223827 |
| hsa-miR-96-5p | 3.462788175 | 1.23E−05 | 0.000801583 |
| hsa-miR-34c-5p | 5.709734189 | 0.000155377 | 0.00499802 |
| hsa-miR-221-3p | 3.083508634 | 0.000447279 | 0.011139374 |
| hsa-miR-1296-5p | 2.77700633 | 0.001063686 | 0.024187298 |
| hsa-miR-4326 | 3.176106886 | 0.002121396 | 0.042672687 |
| **Down regulation** | | | |
| hsa-miR-4686 | −7.902877533 | 5.08E−05 | 0.002416204 |
| hsa-miR-490-3p | −7.688833098 | 0.00016246 | 0.00499802 |
| hsa-miR-552-3p | −5.124795626 | 0.000941615 | 0.022384753 |
| hsa-miR-378i | −2.937299069 | 1.45E−10 | 3.80E−08 |
| hsa-miR-139-5p | −2.765741806 | 2.41E−14 | 1.26E−11 |
| hsa-miR-139-3p | −2.541957974 | 1.69E−05 | 0.000980009 |
| hsa-miR-150-5p | −2.385291852 | 6.39E−05 | 0.002783666 |
| hsa-miR-378f | −2.336952186 | 0.000102422 | 0.003571126 |
| hsa-miR-378d | −2.182500501 | 7.35E−06 | 0.000549497 |
| hsa-miR-378c | −1.909106063 | 2.83E−06 | 0.000246719 |

Note:
Top 10 (up- and downregulated) differentially expressed miRNAs in normal tissues and hepatocellular carcinoma tissues.

## Prognostic analysis of ceRNA-lncRNAs in patients with hepatocellular carcinoma and validation of screened lncRNAs and mRNAs by qRT-PCR

To further evaluate whether the lncRNAs in the ceRNA network have a strong impact on the prognosis of patients with hepatocellular carcinoma, we used Kaplan-Meier curves to analyze the correlation between the expression of the eight lncRNAs in ceRNA and the overall survival of patients with hepatocellular carcinoma. The results demonstrated that three lncRNAs (ZFAS1, FBXL19-AS1 and AC068473.5) were associated with the prognosis of patients with hepatocellular carcinoma (Fig. 6C). The analysis results showed that these three lncRNAs may be potential oncogenes of hepatocellular carcinoma, and their high expression is associated with poor prognosis. Subsequently, we downloaded the clinicopathological parameters and gene expression values of patients with hepatocellular carcinoma from TCGA (Material S7) to perform Cox regression analysis and risk model construction on the screened lncRNAs. The univariate Cox analysis results were consistent with the results of Kaplan-Meier curve analysis, which further indicated that ZFAS1, FBXL19-AS1 and AC068473.5 were prognostic factors in patients with hepatocellular carcinoma ($p < 0.05$; Figs. 6D and S1). Thereafter, three lncRNAs were analyzed by multivariate Cox regression, and the results showed that ZFAS1 ($p < 0.05$) and FBXL19-AS1 ($p < 0.05$) had independent prognostic significance, AC068473.5 ($p > 0.05$) was not

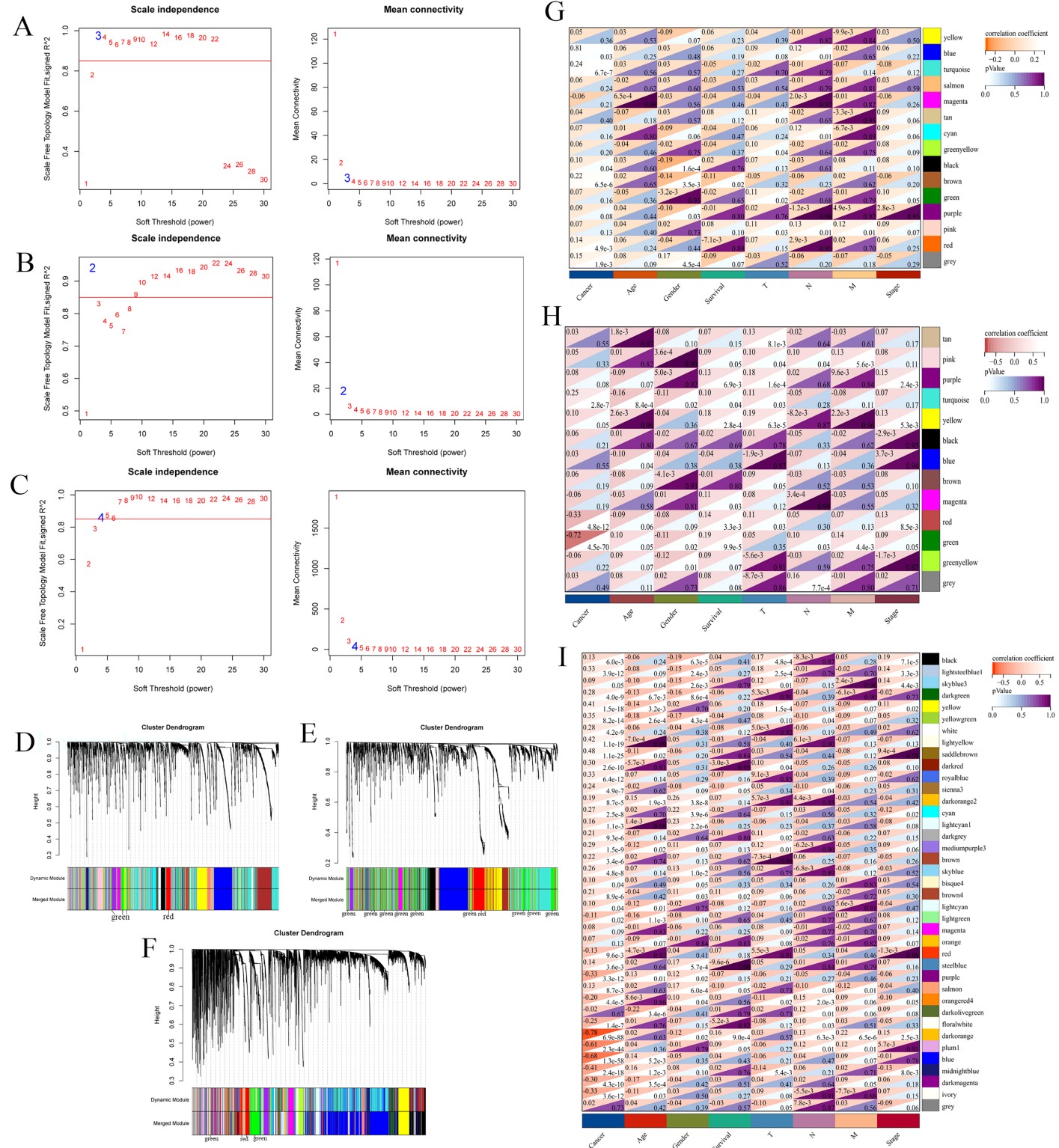

**Figure 3** **WGCNA of RNAs.** Weighted gene coexpression network analysis of RNAs (lncRNAs, miRNAs and mRNAs) in the TCGA database of hepatocellular carcinoma ( TCGA-LIHC ). Scale-free topology criterion and the power β for lnRNA (A), miRNA (B) and mRNA (C). Cluster dendrogram of lncRNAs (D), miRNAs (E) and mRNAs (F) in the coexpression network. The correlations between lncRNA modules (G), miRNA modules (H), mRNA modules (I) and traits are displayed. The light to dark red in the upper left corner represents correlation, and the light to dark purple in the lower right corner represents *p* values.

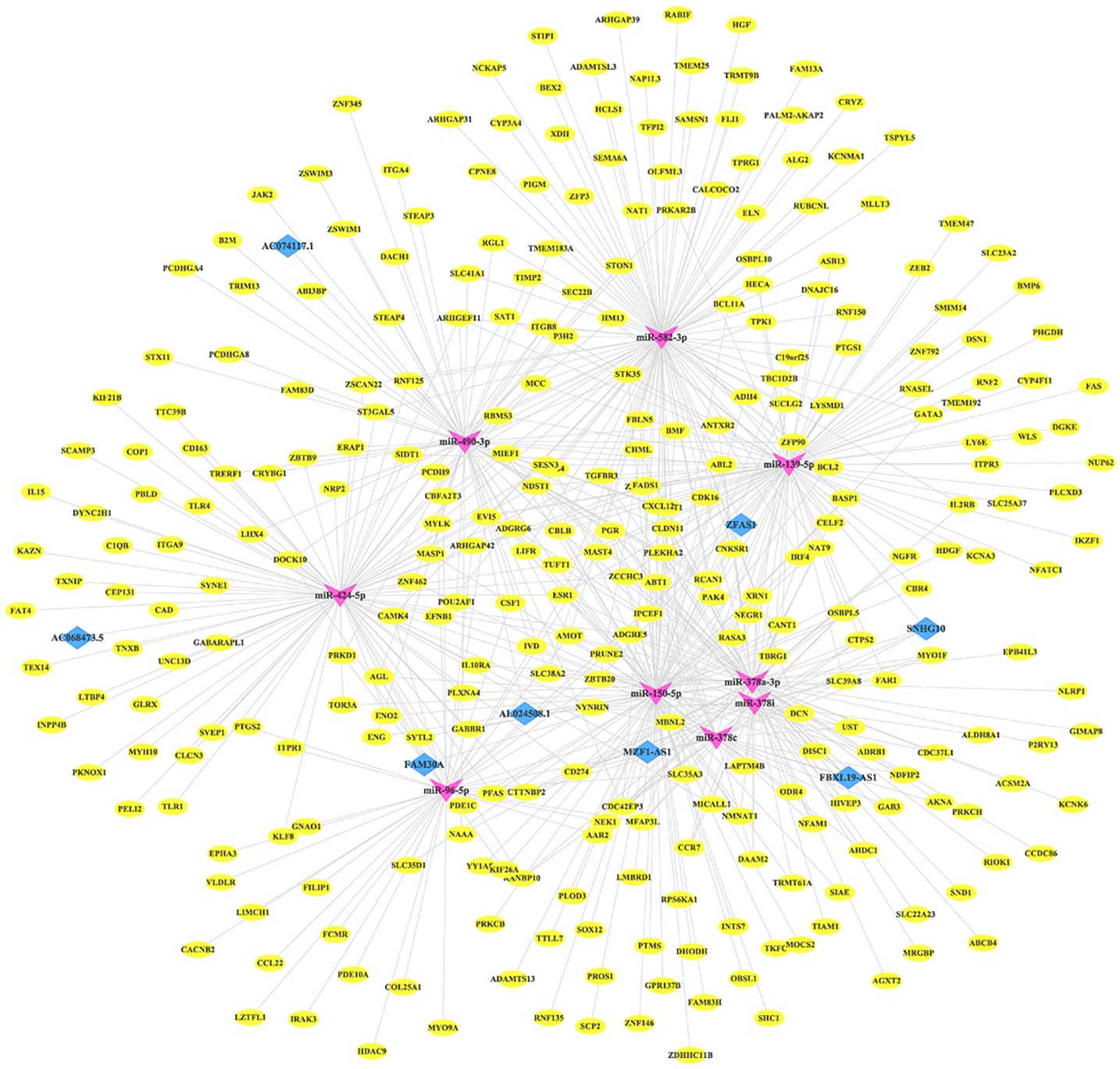

**Figure 4 ceRNA network.** lncRNA-miRNA-mRNA competition endogenous RNA (ceRNA) network of overlapping RNAs between WGCNA and differentially expressed RNAs (lncRNA, miRNA and mRNA). The green arrow represents miRNA, the red diamond represents lncRNA, and the blue oval represents mRNA.

statistically significant (Table 5). Based on the multivariate Cox regression results of ZFAS1 and FBXL19-AS1, we established a risk score model. The calculation formula of the risk score was $(0.002) \times \mathrm{Exp}\ (\mathrm{ZFAS1}) + (0.074)\ \mathrm{Exp} \times (\mathrm{FBXL19\text{-}AS1})$. According to the median risk score, patients with hepatocellular carcinoma were divided into a high-risk

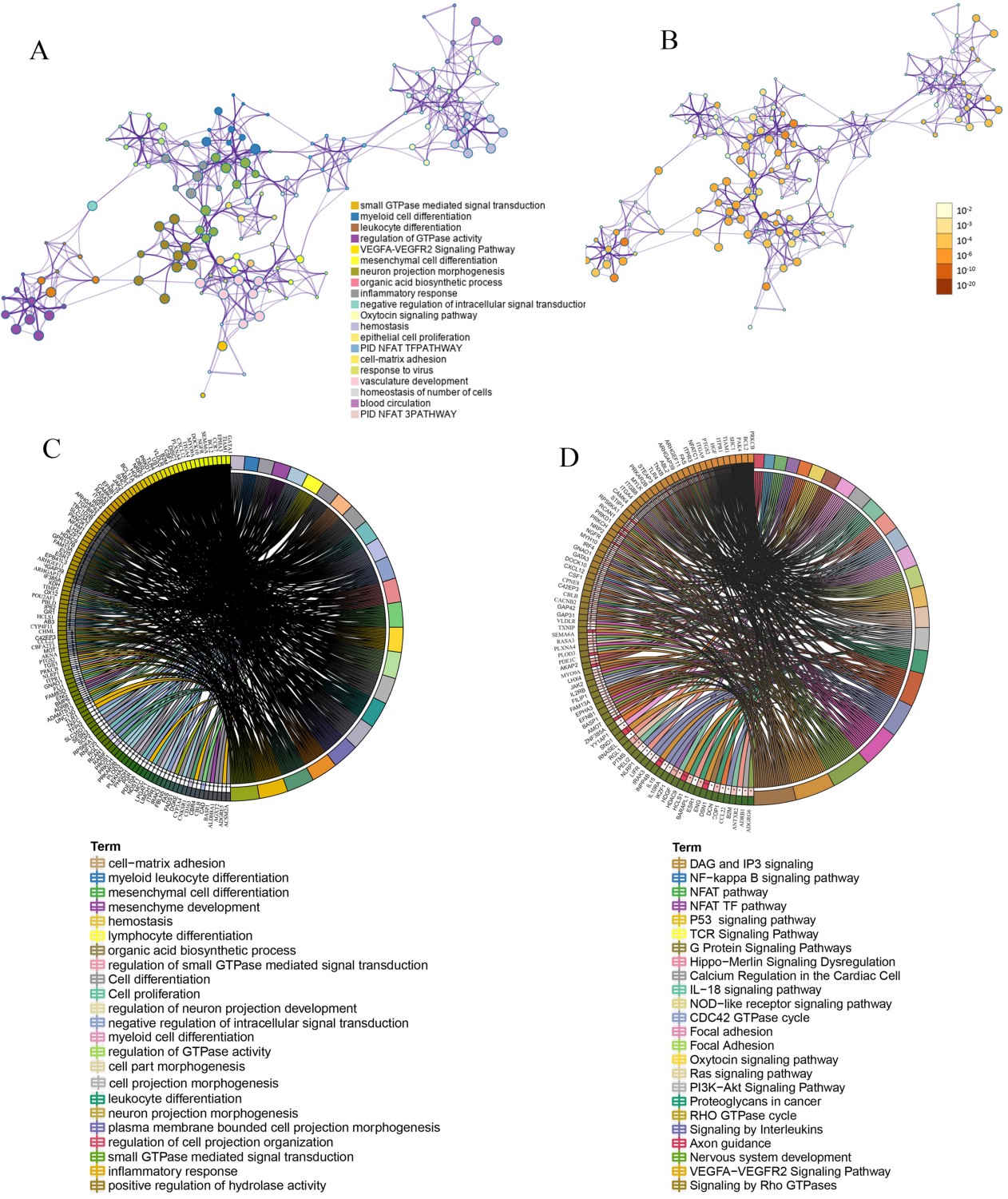

**Figure 5 Function enrichment.** Functional and KEGG enrichment analysis of mRNAs in ceRNA. (A) Network of GO and KEGG enriched terms of mRNAs in ceRNA. A subset of enriched terms was selected and rendered as a network plot, where terms with a similarity >0.3 were connected by edges. a: colored by cluster ID, where nodes that share the same cluster ID are typically close to each other. b: colored by *p* value, where terms containing more genes tend to have a more significant *p* value. (B) Chord diagram showing enriched GO clusters for mRNAs in ceRNA. a: GO enrichment. b: KEGG enrichment. In the chord diagram, mRNAs are shown on the left, and enriched GO clusters are shown on the right.

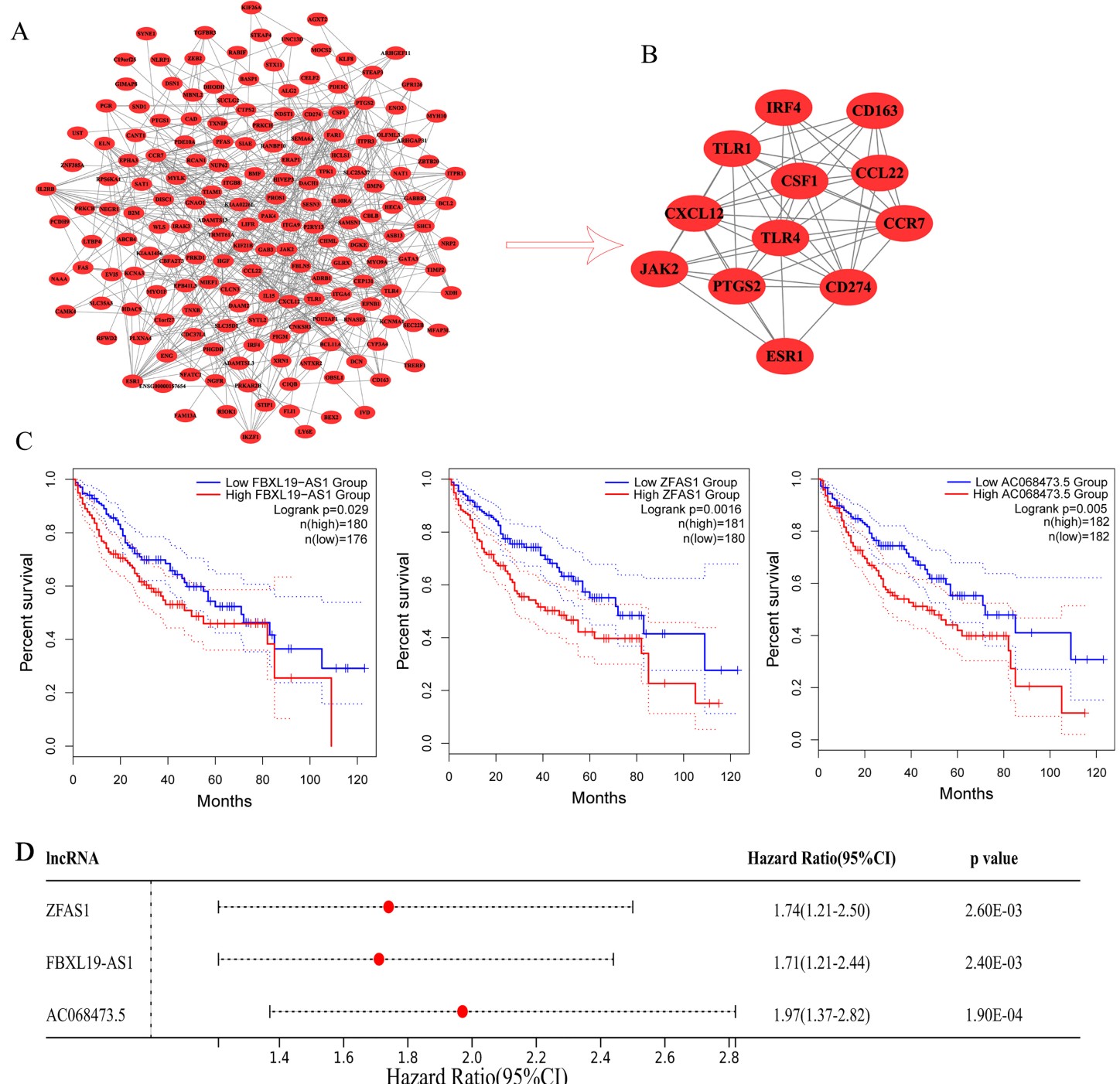

**Figure 6 Prognostic analysis.** (A) PPI networks of the mRNAs in ceRNA. (B) Subnetwork of PPI networks. (C) Kaplan-Meier survival curves for ZFAS1, FBXL19-AS1 and AC068473.5. related to the overall survival of hepatocellular carcinoma patients. (D) The forest plot shows the hazard ratios (HRs) with 95% confidence intervals (95% CI) on the basis of the univariate Cox regression results.

**Table 5 Multivariate cox regression analysis.**

|  | Coefficient | HR | SE | *p*-value |
|---|---|---|---|---|
| ZFAS1 | 1.316 | 1.312 | 0.001 | 4.2E−3 |
| FBXL19-AS1 | 0.074 | 1.002 | 0.028 | 0.020 |
| AC068473.5 | 0.002 | 1.316 | 0.093 | 0.487 |

**Note:**
Multivariate Cox regression analysis of ZFAS1, FBXL19-AS1 and AC068473.5 associated with survival in hepatocellular carcinoma patients.

group and a low-risk group. Risk curve, scatter plot and Kaplan-Meier analyses showed that the overall survival (OS) of the high-risk group was poor (Figs. 7A and 7B). ROC curve analysis showed that the AUCs at 1 year, 3 years and 5 years were 0.78, 0.69, and 0.74, respectively (Fig. 7C). All the results showed that ZFAS1 and FBXL19-AS1 have significant prognostic significance in patients with hepatocellular carcinoma and could be used as independent prognostic factors for hepatocellular carcinoma.

To further verify ZFAS1 and FBXL19-AS1, we used qRT-PCR to analyze the differential expression of lncRNAs in 15 hepatocellular carcinoma tissues and 15 precancerous tissues (Materials S1). The qRT-PCR results showed that the expression of ZFAS1 was consistent with the results of high-throughput sequencing and TCGA data (Figs. 7D and 7E) ($p < 0.05$). There was no statistical significance in the expression of FBXL19-AS1 ($p > 0.05$) between hepatocellular carcinoma tissues and precancerous tissues. Therefore, ZFAS1 is taken as a candidate lncRNA for follow-up research.

## Correlation of the expression of ZFAS1 with clinicopathological parameters and GSEA of ZFAS1

We evaluated the relationship between the expression of ZFAS1 and various clinicopathological parameters in patients with hepatocellular carcinoma. ZFAS1 expression data were obtained from patients with different clinical features of TCGA-LIHC. The results showed that the increased expression of ZFAS1 was significantly correlated with age, TNM stage and clinical stage of hepatocellular carcinoma patients (Fig. 7F; $p < 0.05$). These results indicated that patients with high expression of ZFAS1 were more likely to develop more advanced tumors than patients with low expression of ZFAS1.

GO term and KEGG pathway analyses were used to explore the potential biological function of ZFAS1. GSEA showed that there were significant differences in the enrichment of GO terms and KEGG pathways in the samples with high expression levels of ZFAS1. We selected the signaling pathway with the highest degree of enrichment according to the enrichment score. As shown in Fig. 8A, the biological processes closely related to ZFAS1 were mainly GTPase activity, cell proliferation and differentiation, and immune stress. KEGG was enriched mainly in the oxytocin signaling pathway, PI3K-Akt signaling pathway and JAK stat signaling pathway. These results indicate that the regulation of cell proliferation and differentiation, enzyme activity control and immune regulation are

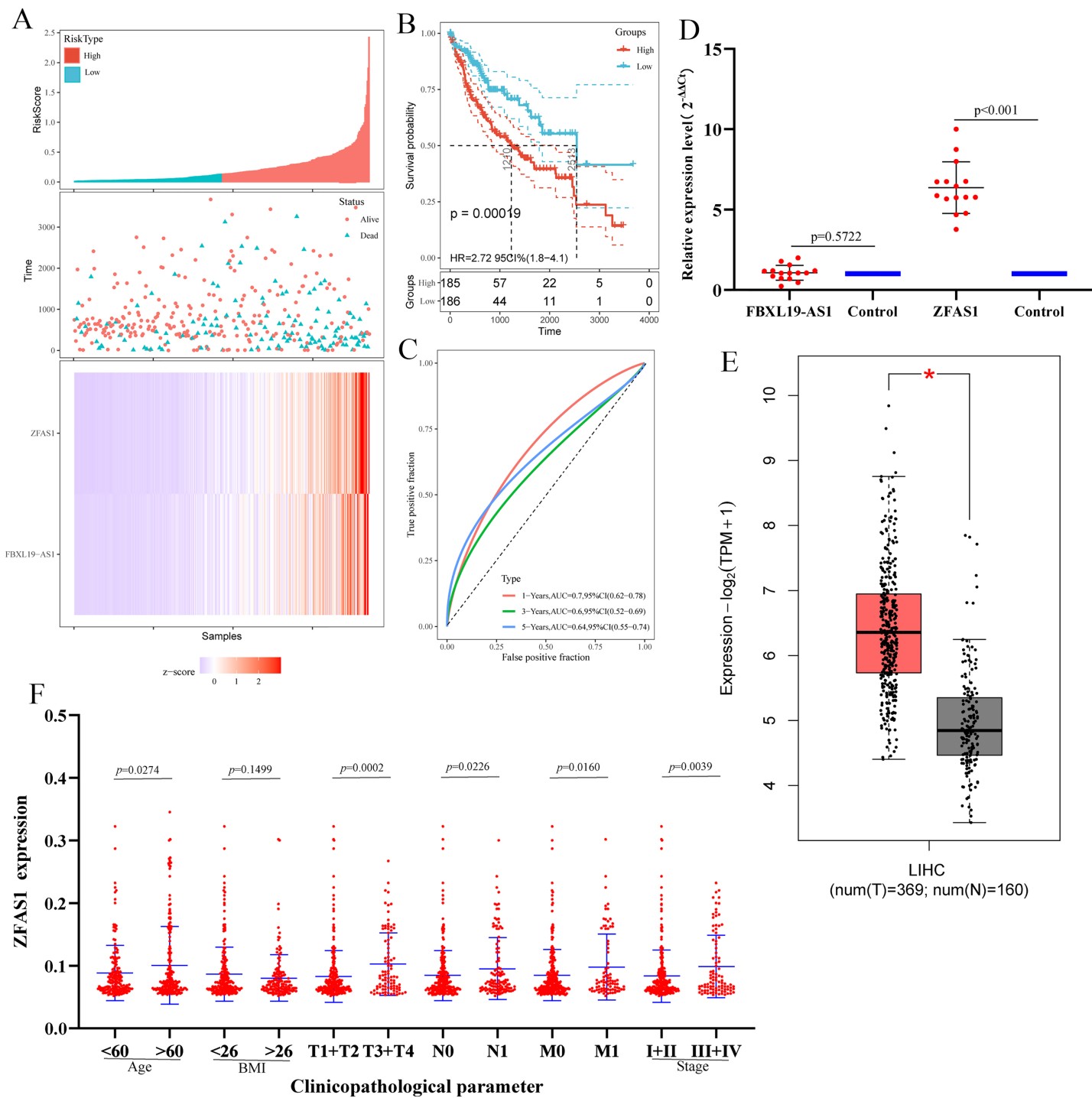

**Figure 7 Risk model construction.** (A) The risk score, scatter plot distribution and heatmap on the basis of ZFAS1 and FBXL19-AS1 at low- and high-risk levels. (B) Kaplan-Meier survival curves of ZFAS1 and FBXL19-AS1 expression at low- and high-risk levels. (C) The ROC curves for predicting survival in hepatocellular carcinoma patients by the risk score. (D) Expression of ZFAS1 and FBXL19-AS1 in hepatocellular carcinoma tissues assessed by qPCR. The expression level of actin was used as an internal reference, and the data are shown as the mean ± standard deviation. (E) Expression of ZFAS1 in hepatocellular carcinoma from TCGA by GEPIA. *$p < 0.05$. "T" represents the tumor, which is represented by the red bar graph; "N" represents the non-tumor, and the blue bar chart indicates. (F) The expression of ZFAS1 correlated significantly with clinicopathological parameters.                             

crucial in hepatocellular carcinoma patients and are closely related to the expression of ZFAS1.

## miR-150-5p was directly regulated by ZFAS1

Subsequently, we predict upstream miRNAs targeting ZFAS1. Venn showed that there were three common miRNAs between the predicted results and the miRNAs in ceRNA: miR-150-5p, miR-582-3p and miR-96-5p (Fig. 8B). miR-582-3p has been investigated as an additional research site in hepatocellular carcinoma in our group. Therefore, miR-150-5p negatively correlated with ZFAS1 expression in the sequencing data and was selected for subsequent study. We subsequently examined miR-150-5p expression in hepatocellular carcinoma tissues. We found that miR-150-5p was significantly decreased in hepatocellular carcinoma tissues compared with adjacent normal tissues (Fig. 8C). Pearson correlation coefficient analysis showed that the clinical expression of miR-150-5p was negatively correlated with the clinical expression of ZFAS1 (Fig. 8D), which was consistent with the sequencing data.

Figure 8E shows the sequence of ZFAS1 3′UTR binding to miR-150-5p. Subsequently, a dual-luciferase assay showed that miR-150-5p could reduce the fluorescence value of transfected ZFAS1-WT cells. The fluorescence value of transfected ZFAS1-MUT cells did not change significantly (Fig. 8F). Furthermore, we found that ZFAS1 silencing could significantly improve the expression level of miR-150-5p by qPCR. The results of overexpression of ZFAS1 were reversed (Fig. 8G). These results indicate that ZFAS1 is directly targeted to miR-150-5p, and the expression of both is negatively correlated.

## ZFAS1 enhanced hepatocellular carcinoma cell proliferation, migration and invasion by targeting miR-150-5p

In this study, we conducted a functional gain-loss assay to investigate the effects of ZFAS1 on the proliferation, migration and invasion of HepG2 cells. The results of the CCK-8 assay showed that overexpression of ZFAS1 promoted the proliferation of HepG2 cells, while silencing ZFAS1 inhibited the proliferation of hepatocellular carcinoma cells (Fig. 9A). The soft agar assay also showed the same trend. Overexpression of ZFAS1 promoted the formation of clones, while inhibition of ZFAS1 expression had the opposite effect (Figs. 9B and 9C). Subsequently, Transwell and wound healing assays showed that the upregulation of ZFAS1 significantly increased cell migration and invasion, and ZFAS1 knockout inhibited the migration and invasion of HepG2 cells (Figs. 9D and 9E). The wound healing rate and invasive cell number of ZFAS1-overexpressing cells were higher than those parameters of the control group, whereas the opposite results were shown in ZFAS1-silenced cells (Figs. 9F and 9G). These results suggested that ZFAS1 could promote the progression of hepatocellular carcinoma cells *in vitro*, which was consistent with the clinical prognosis.

Subsequently, we studied the regulatory relationship between ZFAS1 and miR-150-5p. Function loss assay showed that miR-150-5p inhibitor could reduce the inhibitory effect of si-ZFAS1 on HepG2 cells (Figs. 9A, 9B, 9D and 9E). Similarly, overexpression of miR-150-5p could eliminate the promoting effect of pCDH-ZFAS1 on HepG2 cells (Figs. 9A, 9B, 9D

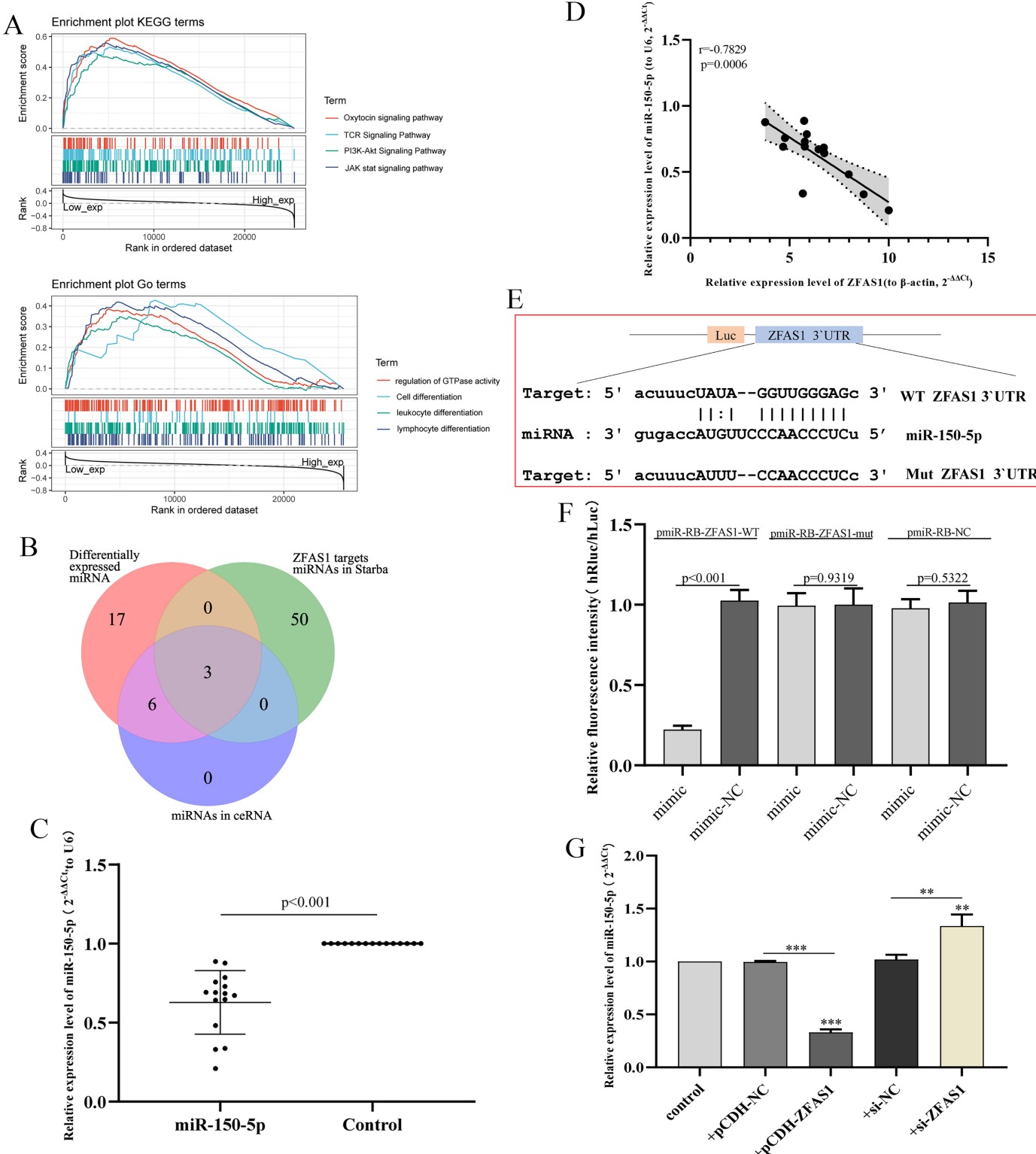

**Figure 8** **Correlation analysis of miR-150-5p and ZFAS1.** (A) Gene set enrichment analysis (GSEA) of ZFAS1. (B) The Venn diagram shows the common miRNAs in the three analysis results. (C) Expression of miR-150-5p in hepatocellular carcinoma tissues assessed by qPCR. The expression level of U6 was used as an internal reference, and the data are shown as the mean ± standard deviation. (D) Correlation analysis between ZFAS1 and miR-150-5p in hepatocellular carcinoma tissues. (E) Sequences of miR-150-5p and the ZFAS1 3′UTR binding sites. (F) Results of the dual-luciferase

**Figure 8** (continued)
assay for miR-150-5p and ZFAS1. (G) Effect of interfering with ZFAS1 expression on miR-150-5p expression in HepG2 cells by qPCR. The expression level of U6 was used as an internal reference, and the data are shown as the mean ± standard deviation. The horizontal line indicates the comparison between groups at both ends, and the nonhorizontal line indicates the comparison with the control group. ***$p < 0.001$. **$p < 0.01$.

and 9E). Together, these findings implied that ZFAS1 promoted the progression of hepatocellular carcinoma cells by regulating miR-150-5p.

# DISCUSSION

lncRNAs have been studied for a long time, and they play vital roles in various biological processes (*Rajagopal et al., 2020*; *Tsagakis et al., 2020*; *Choudhari et al., 2020*; *Gupta et al., 2020*). However, due to the limitations and differences in detection technology and detection platforms, most of the functions of lncRNAs have not been studied systematically. In recent years, abnormal expression of lncRNAs has been found to play a very important role in tumorigenesis and tumor progression. The study of new lncRNAs can also reveal a new mechanism of tumor development (*Sun et al., 2020*; *Olivero & Dimitrova, 2020*; *Zhang et al., 2021b*). Therefore, the exploration of new lncRNAs and the study of their biological functions will provide important impetus for tumor research. Transcriptome sequencing analysis is an effective method for mining new lncRNAs, which can comprehensively analyze tumors (*Ge et al., 2021*). Many studies have shown that the common regulatory mechanism of lncRNA is to act as a miRNA sponge, which is called competitive endogenous RNA (ceRNA), and participate in regulating the expression of target genes. miRNAs can negatively regulate mRNA by binding to complementary sequences. Therefore, lncRNA-miRNA-mRNA interaction has a crucial impact on the development of cancer (*Braga et al., 2020*). More importantly, transcriptome data analysis can also fully identify tumor suppressor and carcinogenic lncRNAs, which is very important to clarify their function and mechanism. At present, although there are many reports about the regulation of hepatocellular carcinoma by lncRNAs, the regulatory mechanism of hepatocellular carcinoma is very complex. In recent years, there have been few reports on new transcripts of hepatocellular carcinoma. To clarify the biological function of more lncRNAs and find new transcripts in hepatocellular carcinoma, we carried out high-throughput sequencing and transcriptome data analysis on three pairs of hepatocellular carcinomas and adjacent tissues.

In this study, we used high-throughput RNA-seq to establish the lncRNA expression profile, mRNA expression profile and miRNA expression profile of HCC and screened 610 lncRNAs, 2,593 mRNAs and 26 miRNAs. To identify important RNAs associated with clinical phenotypes, a lncRNA-miRNA-mRNA ceRNA was constructed in combination with miRNA-targeted lncRNA and mRNA analysis, as well as WGCNA based on TCGA-LIHC data, and eight candidate lncRNAs, nine miRNAs and 307 mRNAs were further obtained. In addition, the 307 hub mRNAs were further analyzed by PPI network and enriched by GO and KEGG analysis. GO analysis was related mainly to cell differentiation, proliferation and immunity, such as leukocyte differentiation, lymphocyte differentiation and cell proliferation. KEGG analysis showed that it was related mainly to

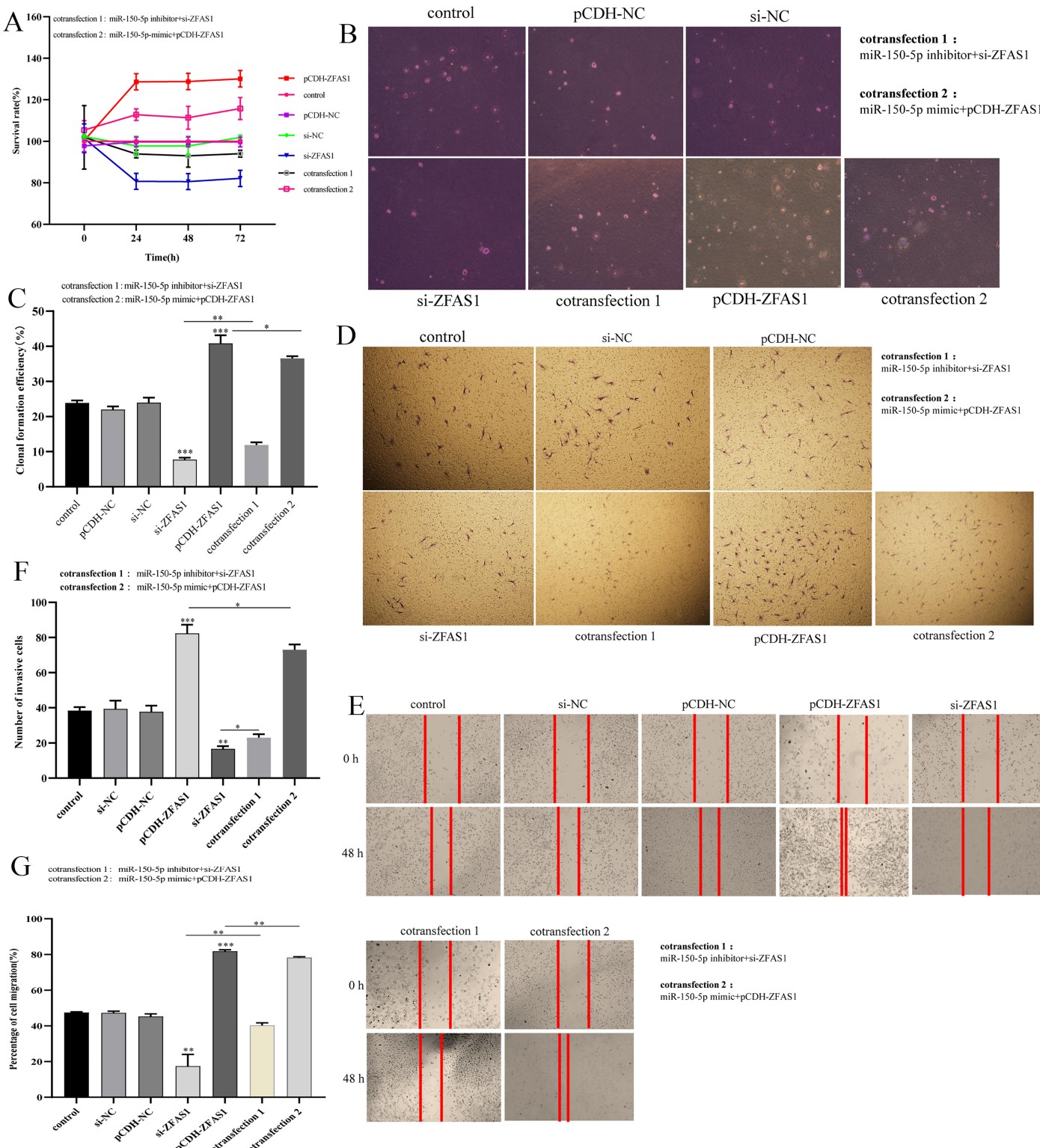

**Figure 9 *In vitro* experiments.** (A) The effects of different experimental groups on the vitality of HepG2 cells measured using the CCK-8 assay. (B) and (C) The effects of different experimental groups on HepG2 cell proliferation measured using a soft agar assay. (D) and (F) The effects of different experimental groups on HepG2 cell invasion measured using a Transwell assay. (E) and (G) The effects of different experimental groups on HepG2 cell migration measured using the wound healing assay. Different experimental groups included control, pCDH-NC, si-NC, si-ZFAS1,

**Figure 9 (continued)**
pCDH-ZFAS1, cotransfection 1 and cotransfection 2. Cotransfection 1: miR-150-5p inhibitor+si-ZFAS1. Transfection 2: miR-150-5p-mimic +pCDH-ZFAS. The horizontal line indicates the comparison between groups at both ends, and the nonhorizontal line indicates the comparison with the control group. NC: negative control. ***$p < 0.001$. **$p < 0.01$. *$p < 0.01$.     

tumor metastasis and immune stress, such as the PI3K-Akt signaling pathway, p53 signaling pathway, G protein signaling pathway, tumor proteoglycan, IL-18 signaling pathway, and interleukin signaling pathway.

In addition, to screen the hub lncRNAs, we combined Kaplan-Meier curve analysis, qPCR, and Cox univariate and multivariate analyses and established a risk model to verify the prognostic potential of lncRNAs. Finally, the lncRNA ZFAS1 with the most significant prognosis was selected. We used the hepatocellular carcinoma data obtained by TCGA to evaluate the prognostic value of ZFAS1 and analyzed the relationship between ZFAS1 and various tumor characteristics, highlighting the role of upregulation of ZFAS1 as an independent prognostic factor of poor OS. Patients with high ZFAS1 expression are more likely to show more advanced grading, staging and tumor status than patients with low ZFAS1 expression. The high expression of ZFAS1 may affect tumorigenesis in the progression of hepatocellular carcinoma. The results showed that the expression of ZFAS1 was closely related to the occurrence and development of hepatocellular carcinoma.

ZFAS1, a novel lncRNA transcribed by the zinc finger antisense orientation of NFX1, is located on human chromosome 20q13.13 (*He et al., 2019*). Initially, ZFAS1 was identified as a new tumor-related lncRNA by upregulating cell proliferation, migration and EMT (*Dong et al., 2018*). In addition, studies have reported that ZFAS1 may play an emerging regulatory role in a variety of diseases, such as acute myocardial infarction (*Zhang et al., 2018*), rheumatoid arthritis (*Zheng et al., 2021*) and cancer (*He et al., 2019*). In cancers, ZFAS1 is thought to be overexpressed in most human cancers, including nonsmall cell lung cancer (*Ge et al., 2020*), colorectal cancer (*Wu et al., 2020*), glioma (*Li et al., 2020*), hepatocellular carcinoma (*Liang et al., 2019*) and head and neck cancer (*Kolenda et al., 2019*). Other studies have shown that the upregulation of ZFAS1 was positively correlated with the clinicopathological features and prognosis of most tumors, such as TNM stage, lymph node metastasis, overall survival and prognosis, and can be used as a biomarker for the prognosis of many kinds of cancer patients (*Romano et al., 2017*; *Lan et al., 2017*).

In hepatocellular carcinoma, ZFAS1 has been identified as a tumor suppressor gene that can affect the progression of hepatocellular carcinoma through various regulatory pathways. *Duan et al. (2020)* reported that ZFAS1 promotes the development of hepatocellular carcinoma through the miR-624/MDK/ERK/JNK/p38 signaling pathway. *Guo et al. (2019)* found that ZFAS1 can promote the migration of hepatocellular carcinoma cells by regulating the production of reactive oxygen species. *Zhou, Zhou & Feng (2019)* reported that ZFAS1 promotes the proliferation of hepatocellular carcinoma by epigenetically inhibiting miR-193a-3p. However, the data of ZFAS1 targeting miR-150-5p regulating hepatocellular carcinoma have not been reported yet, which may be a new mechanism of ZFAS1 regulating hepatocellular carcinoma. We conducted a study on the loss of function of ZFAS1 to explore its effects on the proliferation, migration and invasion of HepG2 cells. We found that si-ZFAS1 can inhibit the proliferation, migration and

invasion of HepG2 cells. In contrast, overexpression of ZFAS1 promotes the proliferation, migration and invasion of HepG2 cells.

Additionally, miR-150-5p has been validated to play important roles in the progression of different types of cancers by regulating lncRNAs. miR-150-5p has been detected to regulate ovarian cancer progression *via* the lncRNA MIAT (*Zhou et al., 2020*). miR-150-5p has also been found to contribute to cell proliferation and migration of colorectal cancer with lncRNA PART1 (*Lou et al., 2020*). lncRNA ZFAS1 targeting miR-150-5p has been reported to be able to promote lung fibroblast-to-myofibroblast transition and ferroptosis (*Yang et al., 2020*). Additionally, lncRNA ZFAS1 promotes ox-LDL-induced EndMT through the miR-150-5p/Notch3 signaling axis (*Yin et al., 2021*). However, little is known about the functional significance of ZFAS1/miR-150-5p in hepatocellular carcinoma, and the potential molecular mechanism of miR-150-5p remains largely unknown in hepatocellular carcinoma. In this study, we demonstrate that ZFAS1 directly interacts with miR-150-5p to influence HepG2 cell proliferation, migration and invasion *in vitro*.

In conclusion, sequencing analysis provided a landscape for the abnormal regulation of lncRNAs and miRNAs in hepatocellular carcinoma. This study presents a novel mechanism of ZFAS1 in the tumorigenesis of hepatocellular carcinoma. We demonstrate that ZFAS1 is increased in hepatocellular carcinoma tissues and correlated with the malignant status and prognosis of hepatocellular carcinoma patients. ZFAS1 promoted hepatocellular carcinoma cell proliferation, migration and invasion *in vitro*. Furthermore, ZFAS1 has also been verified to interact directly with miR-150-5p to influence hepatocellular carcinoma cell proliferation, migration and invasion *in vitro*. However, this model should be verified *in vivo*, and the possible triple axis involved in hepatocellular carcinoma should be investigated.

## ACKNOWLEDGEMENTS

We thank the team that built the GEPIA and StarBase online analysis pages and thank the free online platform of Sanger box tools, and the platform of Shanghai Ordovician Biotechnology Co., Ltd.

### Funding
The authors received no funding for this work.

### Competing Interests
The authors declare that they have no competing interests.

### Author Contributions
- Peng Zhu conceived and designed the experiments, performed the experiments, analyzed the data, prepared figures and/or tables, authored or reviewed drafts of the article, and approved the final draft.

- Yongyan Pei conceived and designed the experiments, performed the experiments, analyzed the data, prepared figures and/or tables, authored or reviewed drafts of the article, and approved the final draft.
- Jian Yu conceived and designed the experiments, analyzed the data, authored or reviewed drafts of the article, and approved the final draft.
- Wenbin Ding performed the experiments, prepared figures and/or tables, and approved the final draft.
- Yun Yang performed the experiments, prepared figures and/or tables, and approved the final draft.
- Fuchen Liu performed the experiments, prepared figures and/or tables, and approved the final draft.
- Lei Liu performed the experiments, prepared figures and/or tables, and approved the final draft.
- Jian Huang performed the experiments, analyzed the data, prepared figures and/or tables, and approved the final draft.
- Shengxian Yuan performed the experiments, analyzed the data, prepared figures and/or tables, and approved the final draft.
- Zongyan Wang analyzed the data, prepared figures and/or tables, authored or reviewed drafts of the article, and approved the final draft.
- Fangming Gu analyzed the data, authored or reviewed drafts of the article, and approved the final draft.
- Zeya Pan analyzed the data, authored or reviewed drafts of the article, and approved the final draft.
- Jinzhong Chen conceived and designed the experiments, performed the experiments, authored or reviewed drafts of the article, and approved the final draft.
- Jinrong Qiu conceived and designed the experiments, authored or reviewed drafts of the article, and approved the final draft.
- Huiying Liu conceived and designed the experiments, analyzed the data, authored or reviewed drafts of the article, and approved the final draft.

### Human Ethics

The following information was supplied relating to ethical approvals (*i.e.*, approving body and any reference numbers):

The Ethics Committee of the Third Affiliated Hospital of Naval Medical University approved the study (EHBHKY2021-K-034).

### Microarray Data Deposition

The following information was supplied regarding the deposition of microarray data:

lncRNA and mRNA sequences are available at GenBank: GSE185799.

miRNA sequences are also available at GenBank: GSE185913.

### Data Availability

The raw measurements are available in the Supplemental Files and at Figshare: Pei, Yongyan (2023): 76604-Raw data.rar. figshare. Dataset. https://doi.org/10.6084/m9.figshare.20748961.v1.

## Supplemental Information

Supplemental information for this article can be found online at http://dx.doi.org/10.7717/peerj.14891#supplemental-information.

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
