# Peer review of "High-throughput sequencing approach for the identification of lncRNA biomarkers in hepatocellular carcinoma and revealing the effect of ZFAS1/miR-150-5p on hepatocellular carcinoma progression"

_PeerJ, doi:10.7717/peerj.14891_

## Round 0.1 · original submission · Minor Revisions

Dear Dr. Liu

Thank you for submitting your manuscript "High-throughput sequencing approach for the identification of lncRNA biomarkers in hepatocellular carcinoma and reveal the effect of ZFAS1/ miR-150-5p on hepatocellular carcinoma progression" to PeerJ. We have now sufficiently received reports from reviewers who find the study interesting. Therefore, after careful consideration, we have decided to invite a minor revision of the manuscript.

As you will see from the reports copied below, the reviewers raise minor concerns regarding some figures representation, statements and grammatical errors that need professional English editing. Therefore, we ask you to address all of the reviewers' comments. Without substantial revisions, we will be unlikely to send the paper back for review.

Important:
If you feel that you are able to comprehensively address the reviewers’ concerns, please provide a point-by-point response to these comments along with your revision. Please show all changes in the manuscript text file with track changes or color highlighting. If you are unable to address specific reviewer requests or find any points invalid, please explain why in the point-by-point response.

Best regards,

Abhishek Tyagi, PhD
Academic Editor

·

Basic reporting

Most of the article is well written, it could be improved in some specific lines for better reading comprehension. Also some figures need to be improved in the size of their fonts.
Line 46 “Transcriptome was performed” is the collection of gene readouts or the collection of all RNA transcripts, in this paragraph it appears to being referred as if it were a technique. Rephrase it.
Line 267 rephrase the beginning of the line it could be as “RNA and miRNA expression profile was downloaded”
Line 279 I suggest to add “negative” before the word correlation
I recommend to add the meaning of ceRNA (competitive endogenous RNA).
Figure 2 lncRNA and mRNA heatmap increase font size, tags cannot being visualized
Figure 6 C it is important to add the resulted p value of your curve analysis.
Every figure is relevant to support each of the findings.
Multiple times in the text the word "dates" is used to refer data. The correct word is "data".

Experimental design

Figure 2 and methodology text does not match (lines 149, 150). DEG and DEM, is there a reason to choose FC cutoff < 2/3? If yes you need to explain why is a difference between text and graphs?
In the same figure miRNA volcano plot, starting point 0 value for x and y axis appears to be not correctly placed, check it.
In the hazard ratio model did you try to use AC068473.5? Then looking its expression via qRT-PCR? It could be good to clarify that in the text.

Validity of the findings

No comment

Reviewer 2 ·

Basic reporting

The article meets the standards of the journal.

Experimental design

The study is rigorous in desigh and logic.

Validity of the findings

no comment

Reviewer 3 ·

Basic reporting

The study is well-structured with novel findings; however, it is not well-written with a lot of grammar mistakes and typos. A lot of the sentences are incomplete, and the manuscript needs professional English editing throughout. The abstract needs to be rewritten in a better coherence by integrating the four sections into a single paragraph.
Some of the sections are highlighted in red for no reason, that needs to be fixed.

Experimental design

The experimental design of the study is sound and well-executed. The authors provided information regarding the deposition of the raw sequencing data.

Validity of the findings

Findings can be used in designing diagnostic and therapeutic approaches in the future.

Additional comments

1. Throughout the figures, the graphics need to be of better quality and the texts/axis labels should be consistent in font and size. This will significantly improve the readability of the manuscript. Some of the axis labels are too small to read…

2. In Figure 7B, the FBXL19-AS1 qPCR showed no statistical significance between control and cancer tissues. What does it mean?

3. In figure 7E, the colors need to be annotated. Which is which? Abbreviations should be explained the first time they come within the manuscript. What is “T” and what is “N”?

4. In figure 8D, what are the units of the X-axis and Y-axis?

---

## Round 0.2 · Minor Revisions

Dear Dr. Pei

Thank you for submitting your manuscript, "High-throughput sequencing approach for the identification of lncRNA biomarkers in hepatocellular carcinoma and revealing the effect of ZFAS1/miR-150-5p on hepatocellular carcinoma progression," to PeerJ. We have now sufficiently received reports from reviewers. Therefore, after careful consideration, we have decided to invite a minor revision of the manuscript.

As you will see from the reports copied below, one reviewer raised concerns regarding some experimental design in the current study. Therefore, we ask you to address the reviewers' concern as appropriate. Without substantial revisions, we will be unlikely to send the paper back for review.

Important:
If you feel that you are able to comprehensively address the reviewers’ concerns, please provide a point-by-point response to these comments along with your revision. Please show all changes in the manuscript text file with track changes or color highlighting. If you are unable to address specific reviewer requests or find any points invalid, please explain why in the point-by-point response.

Best regards,

Abhishek Tyagi, PhD
Academic Editor

·

Basic reporting

The text is clear, it is well written. The figures were improved. English was substantially improved.

Experimental design

There remain one important issue with figure number 2 for miRNA plot. When you have a volcano plot it have the form like "wings" starting from FC = 0, this is because there is a relationship between p-value and fold change, with no fold change (log2FC = 0) you have no significant difference so the p-value is 1 which is then converted to zero after applying the -10log function so when FC starts to increase you can get lower p-values which produces that effect of "wings" after applying -10log function as in figure 2 A lncRNA & mRNA.

The miRNA volcano plot in the log2FC = 0 containts points that have a p-value greater than 1 that means they have some difference which is not posible in that place of the graph. You need to check carefully if its a rendering graph problem or what is happening with that specific graph.

Validity of the findings

No comment

Additional comments

No comment

---

## Round 0.3 · accepted · Accept

Dear Dr. Pei,

We are delighted to accept your manuscript, entitled "High-throughput sequencing approach for the identification of lncRNA biomarkers in hepatocellular carcinoma and revealing the effect of ZFAS1/miR-150-5p on hepatocellular carcinoma progression," for publication in PeerJ.

Thank you for choosing to publish your interesting work with us.


With kind regards,
Abhishek Tyagi
Academic Editor, PeerJ

·

Basic reporting

No comment

Experimental design

No comment

Validity of the findings

No comment

Additional comments

Line 47 has a extra parentheses before the word WGCNA.